# Implant Sizing and Positioning in Anatomical Total Shoulder Arthroplasty Using a Rotator Cuff-Sparing Postero-Inferior Approach

**DOI:** 10.3390/jcm11123324

**Published:** 2022-06-10

**Authors:** Philipp Moroder, Lucca Lacheta, Marvin Minkus, Katrin Karpinski, Frank Uhing, Sheldon De Souza, Michael van der Merwe, Doruk Akgün

**Affiliations:** 1Schulthess Clinic, 8008 Zürich, Switzerland; philipp.moroder@kws.ch; 2Arthrex GMBH, D-81249 Munich, Germany; lucca.lacheta@charite.de (L.L.); marvin.minkus@charite.de (M.M.); katrin.karpinski@charite.de (K.K.); 3Center for Musculoskeletal Surgery, Charité—University Medicine Berlin, Charitéplatz 1, D-10117 Berlin, Germany; frank.uhing@arthrex.de (F.U.); sheldon.desouza@arthrex.de (S.D.S.); michael.vandermerwe@arthrex.de (M.v.d.M.)

**Keywords:** anatomical total shoulder arthroplasty, posteroinferior approach, rotator cuff-sparing, anatomical study

## Abstract

**Background:** The goal of this study was to compare the effectiveness of a rotator cuff-sparing postero-inferior (PI) approach with subdeltoidal access to the traditional subscapularis-takedown deltopectoral approach, in terms of implant sizing and positioning in anatomical total shoulder arthroplasty (aTSA). **Methods:** This study involved 18 human cadaveric shoulders with intact rotator cuffs and no evidence of head deforming osteoarthritis. An Eclipse stemless aTSA (Arthrex, Naples, FL, USA) was implanted in nine randomly selected specimens using a standard subscapularis-tenotomy deltopectoral approach, and in the other nine specimens using the cuff-sparing PI approach. Pre- and postoperative antero-posterior (AP) and axillary fluoroscopic radiographs were analyzed by two independent, blinded raters for the following parameters: (1) anatomic and prosthetic neck-shaft angle (NSA); (2) the shift between the anatomic and prosthetic center of rotation (COR); (3) anatomical size matching of the prosthetic humeral head; (4) the calculated Anatomic Reconstruction Score (ARS); (5) glenoid positioning; as well as (6) glenoid inclination and version. **Results:** While the COR was slightly but significantly positioned (*p* = 0.031) to be more medial in the PI approach group (3.7 ± 3.4%, range: −2.3% to 8.7%) than in the deltopectoral approach group (−0.2 ± 3.6%, range: −6.9% to 4.1%), on average, none of the remaining measured radiographic parameters significantly differed between both groups (PI approach group vs. deltopectoral group: NSA 130° vs. 127°, *p* = 0.57; COR supero-inferior, 2.6% vs. 1.0%, *p* = 0.35; COR antero-posterior, 0.9% vs. 1.7%, *p* = 0.57; head size supero-inferior, 97.3% vs. 98.5%, *p* = 0.15; head size antero-posterior, 101.1% vs. 100.6%, *p* = 0.54; ARS, 8.4 vs. 9.3, *p* = 0.13; glenoid positioning supero-inferior, 49.1% vs. 51.1%, *p* = 0.33; glenoid positioning antero-posterior, 49.3% vs. 50.4%, *p* = 0.23; glenoid inclination, 86° vs. 88°, *p* = 0.27; and glenoid retroversion, 91° vs. 89°, *p* = 0.27). **Conclusions:** A PI approach allows for sufficient exposure and orientation to perform rotator-cuff sparing aTSA with acceptable implant sizing and positioning in cadaveric specimens.

## 1. Introduction

One of the main failure modes after anatomical total shoulder arthroplasty (aTSA) is rotator cuff insufficiency [1]. While reverse total shoulder arthroplasty (rTSA) can function without an intact rotator cuff, aTSA depends on rotator cuff integrity and its ability to center the humeral head on the glenoid, due to the low constraint of the anatomical design itself.

Traditionally, aTSA is performed via an anterior deltopectoral approach that offers good exposure of the humeral head and sufficient exposure of the glenoid via takedown of the subscapularis (SSC). Different types of SSC management are being employed in an effort to simultaneously improve healing and to maintain surgical feasibility at the same time, albeit, without any clear evidence of superiority of one over the other [2,3,4,5]. Regardless of the refixation type, a takedown of the SSC poses a threat to future rotator cuff integrity, and at the same time, warrants postoperative immobilization with the associated discomfort for the patient and risk for stiffness.

Due to these concerns, different types of less cuff-jeopardizing approaches for performing aTSA have been proposed, including an anterior deltopectoral approach with only partial take-down of the inferior subscapularis [6], a superior approach through the deltoid and the rotator interval [7], and an anterior deltopectoral approach through the rotator interval [8]. While the clinical outcome for the complete rotator-cuff sparing interval approaches were comparable to the results obtained with traditional approaches, there was concern regarding non-anatomical neck osteotomies, head sizing, and neck-shaft angle, as well as increased superior decentering and an inability to resect inferior osteophytes in the calcar area, due to limited exposure [7,8,9,10].

Amirthanayagam et al., examined the anatomical feasibility and achievable exposure of the humeral head and glenoid of different anterior and posterior rotator-cuff sparing approaches [11]. They propagated the postero-inferior subdeltoid approach according to Brodsky [12] for implanting an aTSA, because it provides the greatest access while minimizing the damage to the rotator cuff [11]. It seems that anterior cuff-sparing approaches are a trade-off between limited exposure and damage of the crucial anterosuperior aspect of the rotator cuff.

While no clinical reports of aTSA via a posteroinferior subdeltoid approach have been published, Gagey et al., described a posterolateral transdeltoid approach with osteotomy of the external rotators, which allows for a wide exposure that is suitable for primary or revision of total shoulder arthroplasty [13]. Greiwe et al., reported 6-month results for aTSA implanted using a transdeltoid posterior approach with rotator cuff-sparing internervous access to the joint between the infraspinatus and the teres minor, lateral T-shaped capsulotomy, as well as an in-situ osteotomy of the humeral head. The authors conclude that this approach is a safe and effective method for performing aTSA [14]. In an anatomical study of the same approach, on average, 89% of the glenoid and 95% of the humeral cut surface could be visualized, and the stem could be reliably implanted in neutral angulation. However, the authors also stress the point that it is a challenging technique that should not be attempted in clinical practice without proper training [15].

In this anatomical study, we explored the possibility of a rotator cuff-sparing implantation of an anatomical total shoulder arthroplasty (aTSA) via the postero-inferior (PI) approach with subdeltoidal access and the posterior dislocation of the humeral head through an internervous split between the infraspinatus and the teres minor, for improved exposure and the precise osteotomy of the humeral head. The goal of this study was to compare the effectiveness of this PI approach to the traditional subscapularis takedown deltopectoral approach, in terms of sizing and positioning, when implanting an aTSA.

## 2. Materials and Methods

Prior to the beginning of this study, institutional ethical committee approval was obtained (EA1/026/21). 20 fresh frozen right-sided cadaveric shoulders (Science Care Inc., Phoenix, AZ, USA) were obtained. Only specimens with intact rotator cuffs and no evidence of head deforming osteoarthritis were employed for this study, leaving 18 shoulders from 13 female and 5 male donors, with a mean age at the time of death of 72 years (range: 56–93 years), for further evaluation. The cadavers were randomly divided into 2 groups of 9 specimen each. An Eclipse stemless anatomical shoulder arthroplasty (Arthrex, Naples, FL, USA) was implanted in the first group, using a standard deltopectoral approach, and in the second group, using a PI approach. True anteroposterior (AP) and axillary fluoroscopic radiographs were obtained from each specimen, pre- and postoperatively. On the postoperative images, care was taken to gather perfectly orthogonal images of the arthroplasty without any overlap between the metallic trunion and the bony surface of the osteotomy.

### 2.1. Surgical Technique

All surgeries were performed by the first author (PM), who was assisted by the senior author (DA).

#### 2.1.1. Deltopectoral Approach

In the first group, a standard deltopectoral approach with subscapularis tenotomy was performed. After dislocation of the humeral head, resection was obtained at the level of the anatomic neck. The trunion size was then determined using a template, and the length of the cage screw was measured using a special cage screw sizer. Then, a complete exposure of the glenoid articular surface was obtained by releasing the labrum and the capsule. The size of the glenoid (small, medium, or large) was assessed by choosing the glenoid guide that best matched the glenoid surface area. The guide pin was then inserted through the guide into the glenoid vault until it reached the medial cortical bone. After removal of the guide, the inserted pin was cut at the level of the glenoid surface, leaving the rest of the pin in the glenoid vault to allow for later radiological evaluation of the appropriate pin positioning, as well as the pin version and inclination. No glenoid implantation was performed; however, prior to cutting the pin, a reamer was inserted in order to evaluate the theoretical level of difficulty to ream the glenoid without actually reaming the bony surface. The definitive trunion was then implanted onto the resection plane, and the definitive cage screw was inserted followed by the appropriate size humeral head. The sizing and positioning of the implant was performed in a manner representative of the current standard of care in clinical practice.

#### 2.1.2. Posterior Approach

A subdeltoid approach, previously described by Brodsky et al. [12], was used in all cases. A 10–12 cm vertical skin incision was made on the posterior aspect of the shoulder, beginning at the posterior border of the acromion around 2 cm medial of the lateral aspect and extending inferiorly slightly lateral to the posterior axillary fold. After identifying and mobilizing the inferior border of the spinal part of the deltoid, the deltoid was retracted superiorly and laterally facilitated by the abduction of the arm. No splitting of the deltoid muscle was performed. Next, the internervous interval between the infraspinatus and the teres minor was visually identified, and its distance to the axillary nerve was measured and documented (Figure 1). A fat line between the infraspinatus and teres minor could be identified in two-thirds of the specimen to aid in identifying the internervous interval. A split between the teres minor and infraspinatus was performed with a subsequent lateral “T-shaped” capsular incision in the first three consecutive cases, and a medial “T-shaped” capsular incision in the subsequent six cases, as the latter offered a better visualization and exposure of the humerus and the glenoid. The humeral head was then dislocated posteroinferiorly through the created interval via flexion, horizontal adduction, and internal rotation (Figure 2). The humeral cut was performed at the level of the anatomic neck, while carefully protecting the rotator cuff with retractors. The trunion size was then determined using a template, and the length of the cage screw was measured using a cage screw sizer. Next, the glenoid was exposed, the labrum excised, and the capsule released around the glenoid (Figure 3). The size of the glenoid was then measured, and the guide pin was inserted and cut at the level of the glenoid surface, as previously described for the deltopectoral approach, along with the simulated glenoid reaming without actually removing bone.

While performing the procedure, the surgeon had to grade the difficulty (poor, acceptable, or excellent) to achieve a certain surgical step, including identification of the internervous interval between infraspinatus and teres minor, exposure of the humeral head, humeral head resection, and exposure of the glenoid, as well as glenoid reaming. At the end of the surgery, the surgeon’s satisfaction score (0–100%) regarding the overall surgery process was noted for each case prior to taking the postoperative radiographs and revealing the quality of the implant sizing and placement.

### 2.2. Radiographic Analysis

Radiographic analysis was performed by two independent reviewers (LC and MM) who were not involved in the surgeries performed, and who were blinded to which case had performed, utilizing a deltopectoral or a PI approach. All measurements were performed with the image analysis software Visage 7.1 (Visage Imaging, Berlin, Germany). The preoperative and postoperative neck-shaft angle (NSA) was determined on pre- and postoperative AP radiographs, as previously described by Flurin et al. [16] (Figure 4). For the determination of the native center of rotation (COR) on the postoperative AP radiographs, a best-fit circle was placed based on three preserved bony landmarks, as previously described [17,18]: (1) the lateral cortex of the greater tuberosity; (2) the medial calcar at the inflection point; and (3) the medial footprint of the rotator cuff on the greater tuberosity. A second implant-matched circle was then placed to fit the curvature of the prosthetic humeral head. The COR from each circle was then identified, and a coordinate system was generated from the native COR, with the y-axis aligned parallel to the intramedullary axis of the shaft, and the x-axis as perpendicular to the shaft. The deviation between the pre- and postoperative COR was then determined in the x- and y-axis [18]. In the x-coordinate plane, a shift of postoperative COR was medially considered as positive, and a shift laterally negative, while in the y-coordinate plane, a superior shift was considered as positive, and an inferior shift, negative. The measured distance between the native and the postoperative COR in the medio-lateral and the supero-inferior directions were then each divided by the diameter of the native best-fit circle, and were reported as a percentage (Figure 5a). In addition, the shift between the pre- and postoperative COR in anteroposterior direction was determined on axillary radiographs in a similar fashion. A best-fit circle was fitted on the two edges of the humeral resection plane, with its COR corresponding to the middle of the resection plane. A second implant matched circle was then placed to fit the curvature of the prosthetic humeral head. The COR was then identified from each circle, and a coordinate system was generated from the anatomic COR, with the y-axis aligned parallel to the intramedullary axis of the shaft, and the x-axis perpendicular to this line, to measure the distance between both COR in the anteroposterior direction. In the x-coordinate plane, a shift of postoperative COR anterior was considered positive, and a shift posterior, negative. The measured distance between the native and postoperative COR in an anteroposterior direction was then divided by the diameter of the native best-fit circle, and was given as a percentage (Figure 5b).

Furthermore, the supero-inferior and antero-posterior size matching between the humeral resection plane and prosthetic humeral head diameter were determined and expressed as percentages by dividing the prosthetic head diameter by the length of the humeral resection plane in the AP and axillary radiographs, respectively (Figure 5c,d).

Each radiographic parameter obtained for the humeral component was rated based on the scoring system described in Table 1. The single scores were then summed to yield the anatomic reconstruction score (ARS) for each case, to objectively quantify and to assess the quality of the anatomical humeral head reconstruction.

The pin positioning for the glenoid preparation was assessed in the supero-inferior and the antero-posterior directions by dividing the distance between the pin and inferior glenoid rim by the length of the glenoid, as well as the distance between the pin and the posterior glenoid rim and the width of the glenoid on the AP and the axillary radiographs, respectively (Figure 6a,b). These values were then displayed as percentages. In addition, the inclination and the version of the theoretical glenoid implantation were measured by determining the angle between the native glenoid surface and the glenoid guide pin on the AP and axillary radiographs, respectively (Figure 7a,b).

### 2.3. Statistical Analysis

Intraclass correlation coefficients (ICC) with a 95% confidence interval (CI) were calculated for all measurements. As recommended by Landis and Koch, an ICC < 0.20 resembles slight agreement, 0.21 to 0.40, fair agreement, 0.41 to 0.60, moderate agreement, 0.61 to 0.80, substantial agreement, and >0.81, almost perfect agreement [19]. After a reliability assessment, the values of both raters were averaged for further analysis. The Kolmogorov–Smirnov test was used to test for normal distribution. The two-sample *t*-test (for parametric distribution) or the Mann–Whitney U test (for nonparametric distribution) were used to compare continuous variables between groups. For statistical analyses, IBM SPSS Statistics 25.0 software (IBM, Armonk, NY, USA) was employed. A *p*-value < 0.05 was considered significant.

## 3. Results

The average head and trunion size in the deltopectoral group was 42, with a range from 39 to 47; and the mean size in the PI group was 41, with a range from 39 to 43. In the deltopectoral group, six small, two medium, and one large screw were implanted, while in the PI group, three small, four medium, and two large screws were used. In both groups, five small, three medium, and one large glenoid component were trialed.

While performing the PI approach, the identification of the internervous plane was poor in four cases, acceptable in three cases, and excellent in two cases. The average distance of the internervous plane to the axillary nerve was 33 mm, with a minimum distance of 25 mm and a maximum distance of 45 mm. The posteroinferior dislocation of the humeral head through the internervous interval and inferior to the posterior deltoid was possible in all cases. The exposure of the humeral head was excellent in five cases, acceptable in three cases, and poor in one case. The possibility of resection of the humeral head was excellent in three cases, acceptable in three cases, and poor in three cases. In all cases, with a medial T-shaped incision of the capsule instead of a lateral T-shaped incision, an acceptable or excellent exposure and resection opportunity were identified. The exposure of the glenoid and simulated reaming was excellent in four cases, acceptable in three cases, and poor in two cases. The surgeon satisfaction rating with the procedure when performing the PI approach displayed a learning curve with a positive impact of the switch from a lateral T-shaped to a medial T-shaped incision of the capsule between cases 3 and 4 (Figure 8).

The intraclass correlation coefficients for the radiographic measurements of the two independent observers was almost perfect for six parameters, substantial for two parameters, moderate for two parameters, and fair for two parameters (Table 2).

While the COR was positioned significantly (*p* = 0.031) more medial in the PI approach group (3.7 ± 3.4%, range: −2.3% to 8.7%) than in the deltopectoral approach group (−0.2 ± 3.6%, range: −6.9% to 4.1%) on average, none of the remaining measured radiographic parameters were significantly different between both groups (Table 3).

## 4. Discussion

The investigated PI approach involves a subdeltoidal access and an internervous split between the infraspinatus and the teres minor, as described by Brodsky et al. in 1987 [12]. Furthermore, it includes a medial T-shaped incision of the capsule with iuxtaglenoidal posterior to inferior capsular release to allow for the posteroinferior dislocation of the humeral head, and thus extended exposure for precise humeral head osteotomy. The study results show that the implantation of an anatomical total shoulder arthroplasty with acceptable implant sizing and positioning can be performed via a PI approach in a cadaveric shoulder.

While no statistically significant difference was observed regarding the neck-shaft angle, the variation and range seemed to be a little wider in the PI approach group, with a tendency towards more valgus positioning of the head in some cases. This can be explained by the fact that after postero-inferior dislocation of the humeral head through the internervous interval, the calcar tends to be covered by the teres minor, which needs to be pushed inferiorly, while the superior insertion of the rotator cuff is more easily exposed (Figure 2).

The COR was slightly but statistically significantly more medial in cases with the PI approach than with the deltopectoral approach, indicating a risk of lateral overstuffing due to insufficient resection of the humeral head. While this could be explained by a lack of exposure, it may also be caused by the presence of the bare area on the posterior side of the humerus, which makes identification of the anatomical neck more difficult. Since the neck-shaft angle is above 90°, a lack of resection of the anatomical neck tendentially also leads to a superior translation of the COR, which however, was only slightly observable in this study, and did not yield statistically significant differences. The anteroposterior positioning of the COR showed no difference between the groups.

No differences in terms of sizing of the prosthetic head were observed, with slight supero-inferior undersizing but good antero-posterior matching in both groups, as the resection plane is usually oval shaped with a smaller antero-posterior than supero-inferior diameter [20]. The larger variation in the PI approach group is likely explained by the learning curve.

While the Anatomic Reconstruction Score was not statistically different in both groups, there was a trend towards slightly lower scores in the PI group, mostly explained by the larger variation of the neck-shaft angle and the lack of sufficient resection of the humeral head.

Due to the anterior tilt of the scapula, the described approach offers a postero-inferior direct view of the glenoid, which can be changed to an e-face view when the humeral head is retracted anteriorly. This may lead to a tendency of postero-inferior placement and an increased retroversion of the glenoid guide pin in cases with poor exposure. Greiwe et al. point at different advantages of the posterior approach, including easier access to the retroverted glenoids, as well as facilitated posterior soft tissue balancing [14]. However, it remains unclear as to whether posterior approaches may also weaken the posterior soft tissues, including the posterior capsule and rotator cuff, and therefore, this may possibly aggravate the posterior humeral subluxation in patients with posterior eccentric glenoid wear.

Posterior approaches have already been used in the clinical setting to implant shoulder arthroplasties. Gagey et al., were able to implant 53 hemiarthroplasties through a posterolateral approach with subperiosteal detachment of the posterior part of the deltoid muscle and osteotomy of the external rotator muscles [13]. Although this approach provides a wide range of exposure of the glenoid and humeral head, deltoid release and detachment of the external rotators by means of an osteotomy are the main limitations, as they warrant postoperative immobilization of the arm and pose the risk for deltoid atrophy [13] and the insufficiency of external rotators. In contrast, the PI approach used in this study spares the deltoid and external rotators, and therefore, it allows for immediate postoperative mobilization. Greiwe et al., performed a total shoulder arthroplasty in 31 patients using a posterior rotator cuff-sparing approach, which uses a split of the middle and posterior heads of the deltoid muscle, a lateral based T-shaped capsular incision, and an in-situ humeral osteotomy [14]. Short-term follow up was available for 26 patients, with a significant improvement in clinical scores.

The authors also conducted an anatomic feasibility study to evaluate their approach, and this showed good access to glenoid and humerus, despite the mentioned technical difficulties [15]. While the deltoid split does not seem to affect deltoid integrity [14], the in-situ osteotomy of the humeral head, which is performed without dislocation of the head and via the internervous split, poses a surgical challenge, due to limited exposure and few bony landmarks for reference. As the identification of the anatomic neck for a precise humeral osteotomy may be difficult, there is a risk for an improper humeral cut, which can lead to malpositioning of the prosthetic humeral head, and potentially cause asymmetric loading of the glenoid, resulting in glenoid erosion and loosening [21,22,23]. According to the present anatomical study, a medial T-shaped incision of the capsule, with iuxtaglenoidal posterior to inferior capsular release, instead of a lateral T-shaped incision, may facilitate posteroinferior dislocation of the humeral head through the internervous interval, and thus allow for a precise identification of the anatomic neck, and easier humeral head osteotomy. However, great attention must be given not to stretch and harm the axillary nerve with the retractors placed inferiorly between the dislocated humeral head and the teres minor. Finally, it must be mentioned that even though the step of posterior dislocation of the humeral head can quite easily be obtained in cadaveric specimens, it might not be achievable in patients with severe joint stiffness due to advanced osteoarthritis.

A limitation of this study is the fact that the implantation of arthroplasties in cadaveric shoulders is typically easier, due to the reduced tension of the soft tissues. This might have facilitated the exposure and the implantation, especially in the PI approach group, as even in the cadaveric setting, only limited exposure could be obtained in some cases. Furthermore, not whole-body, but rather mere shoulder specimens were used for this study, making the placement easier to handle than what could be expected in clinical practice. While most measurement parameters have proven to be reliable, with acceptable ICCs, two parameters (pre-operative neck-shaft angle and the antero-posterior COR) showed only fair ICCs, thus limiting their interpretabilities. Finally, no conclusions regarding the risk of damage to the axillary nerve when performing the PI approach can be drawn from this study. Despite the apparently sufficient distance to the interval between the teres minor and the infraspinatus, no information on the changes in position and tension on the nerve during the posterior dislocation of the humeral head, humeral and glenoid exposure, as well as motion of the arm were collected.

## 5. Conclusions

The investigated postero-inferior approach with subdeltoidal access and posterior dislocation of the humeral head through an internervous split between the infraspinatus and the teres minor with a medial T-shaped incision of the capsule allows for sufficient exposure and orientation to perform rotator-cuff sparing anatomical total shoulder arthroplasty with acceptable implant sizing and positioning in cadaveric specimens. This approach tends to medialize the COR that needs to be taken into account when performing aTSA. Further research should focus on the radiological and clinical outcomes of the PI approach in daily practice.

## Figures and Tables

**Figure 1 jcm-11-03324-f001:**
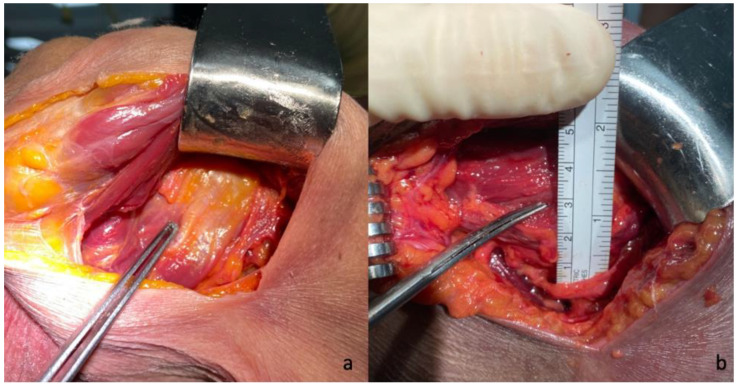
Identification of the internervous interval between the infraspinatus and the teres minor (**a**) and measurement of its distance to axillary nerve (**b**) in a postero-inferior approach.

**Figure 2 jcm-11-03324-f002:**
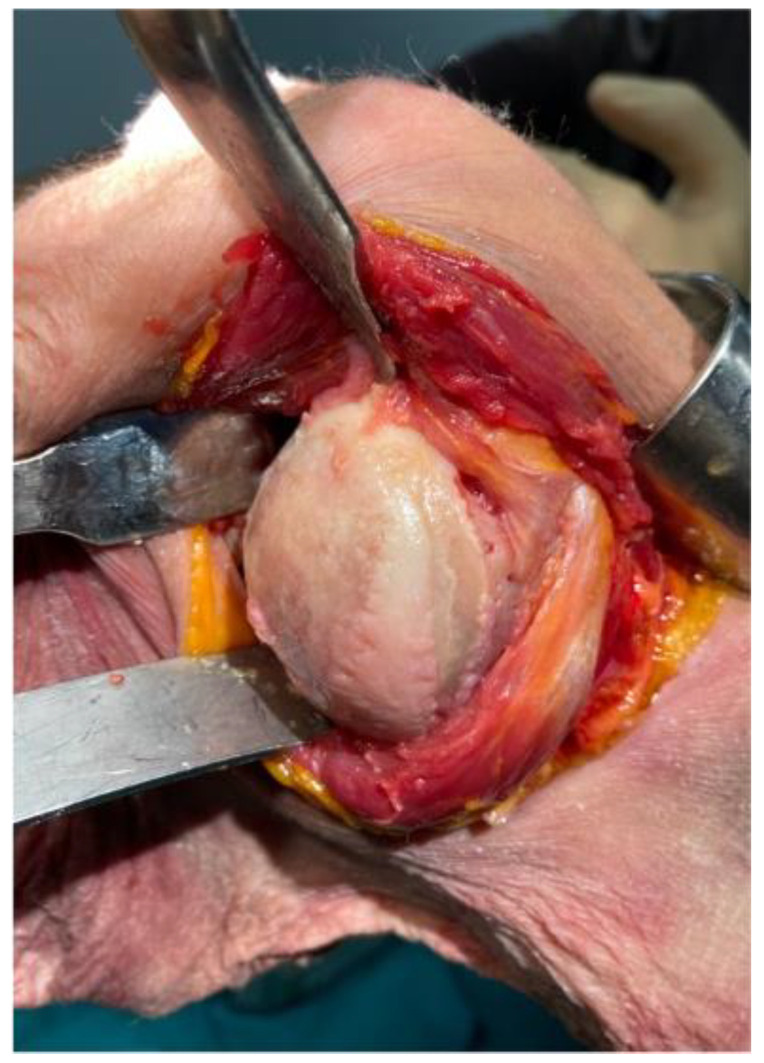
Humeral head exposure via the postero-inferior approach after posteroinferior dislocation through the internervous interval and below the deltoid muscle.

**Figure 3 jcm-11-03324-f003:**
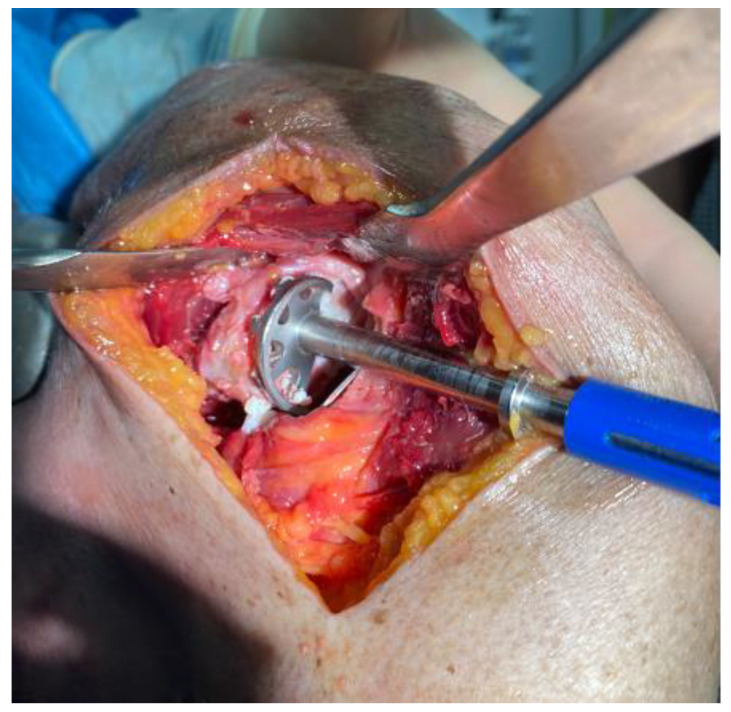
Glenoid exposure and simulated reaming through the postero-inferior approach.

**Figure 4 jcm-11-03324-f004:**
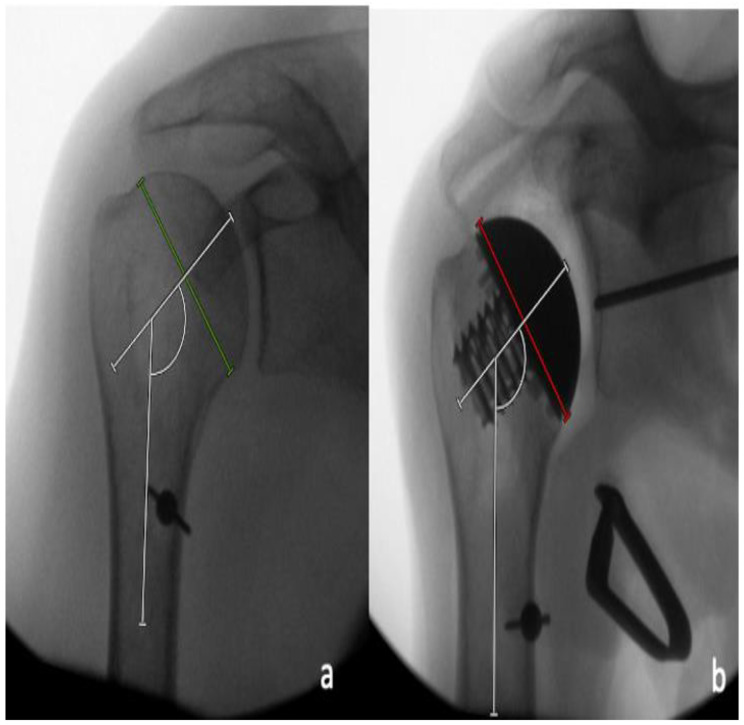
(**a**) Determination of the pre-operative neck-shaft angle between the line perpendicular to the anatomic neck axis (green line) and the intramedullary axis. (**b**) Determination of the post-operative neck-shaft angle between the line perpendicular to the backsurface of the trunion (red line) and the intramedullary axis.

**Figure 5 jcm-11-03324-f005:**
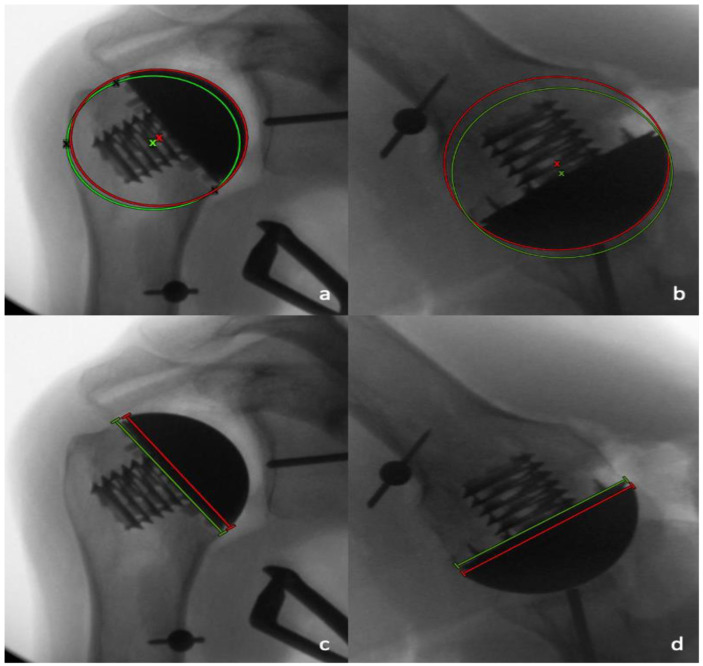
A best-fit anatomic circle (green circle) with its center of rotation (COR) (green x), and a best-fit implant circle (red circle) with its COR (red x) are placed on the AP (**a**) and axillary (**b**) views to determine the differences in the positioning of the COR. The length of the resection plane (green line) and the prosthetic humeral head (red line) were compared on the AP radiographs (**c**) and the axillary radiographs (**d**).

**Figure 6 jcm-11-03324-f006:**
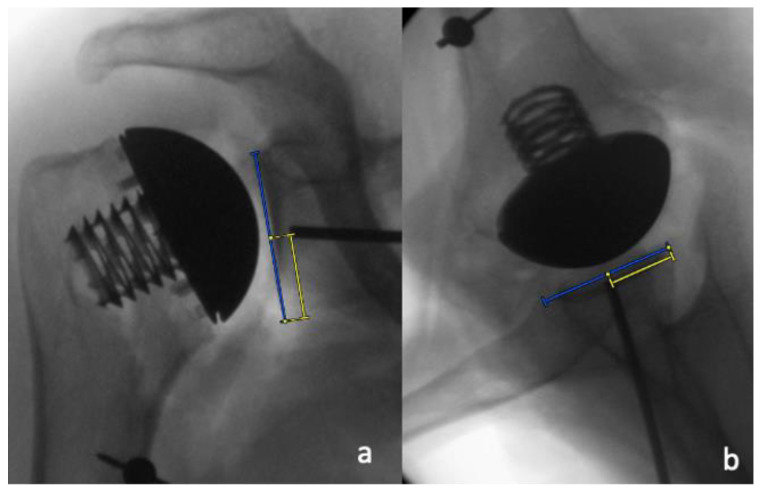
Pin placement was determined (**a**) by dividing the distance of the pin to the inferior glenoid rim (yellow line) by the supero-inferior extent of the glenoid (blue line) on the AP radiographs, and (**b**) the distance of the pin to the posterior glenoid rim (yellow line) by the antero-posterior extent of the glenoid (blue line) on the axillary radiographs.

**Figure 7 jcm-11-03324-f007:**
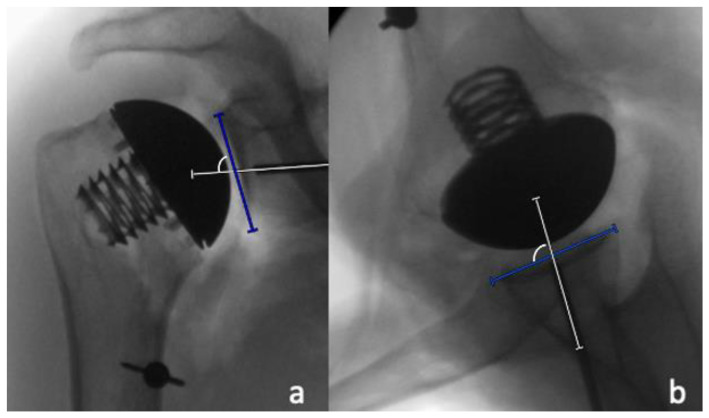
Measurements of theoretical glenoid component inclination (**a**) and version (**b**) by determining the angle between the native glenoid surface and the glenoid guide pin. A guide pin placed with more inclination or retroversion leads to a larger recorded angle.

**Figure 8 jcm-11-03324-f008:**
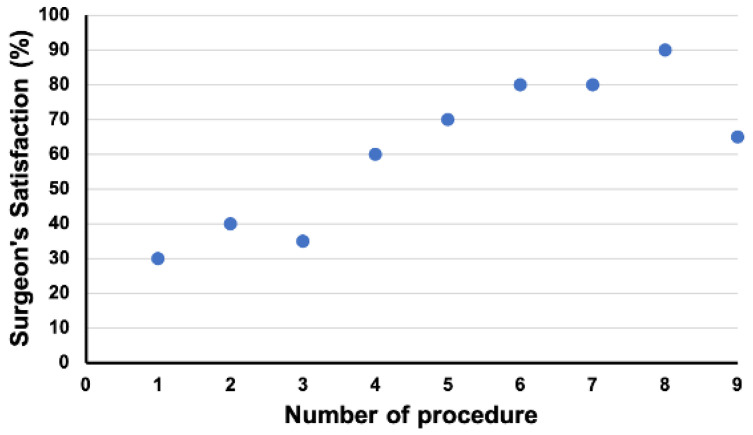
Change in surgeon’s satisfaction from the first case to the ninth case when performing total shoulder arthroplasty via a posteroinferior approach. Between cases 3 and 4, a switch from a lateral T-shaped to a medial T-shaped incision of the capsule was made.

**Table 1 jcm-11-03324-t001:** Parameters used to calculate the Anatomic Reconstruction Score (ARS).

Neck-Shaft Angle	Pre and Post Difference
Rating of 0 points	>10°
Rating of 1 points	>5° and ≤10°
Rating of 2 points	≤5°
COR medio-lateral	Pre and post difference
Rating of 0 points	>10%
Rating of 1 points	>5% and ≤10%
Rating of 2 points	≤5%
COR supero-inferior	Pre and post difference
Rating of 0 points	>10%
Rating of 1 points	>5% and ≤10%
Rating of 2 points	≤5%
COR antero-posterior	Pre- and postoperative difference
Rating of 0 points	>10%
Rating of 1 points	>5% and ≤10%
Rating of 2 points	≤5%
Head size supero-inferior	
Rating of 0 points	<95% or >105%
Rating of 1 points	≥95% and ≤105%
Head size antero-posterior	
Rating of 0 points	<95% or >105%
Rating of 1 points	≥95% and ≤105%

COR, center of rotation.

**Table 2 jcm-11-03324-t002:** Intra-class correlation coefficients (ICC) of the radiographic measurement parameters.

Measurement Parameter	ICC
Neck-Shaft Angle native (°)	0.289
Neck-Shaft Angle post-operative (°)	0.961
COR medio-lateral (%)	0.823
COR supero-inferior (%)	0.484
COR antero-posterior (%)	0.303
Head size supero-inferior (%)	0.765
Head size antero-posterior (%)	0.898
Glenoid positioning supero-inferior (%)	0.868
Glenoid positioning antero-posterior (%)	0.506
Glenoid Inclination (°)	0.938
Glenoid Retroversion (°)	0.980

**Table 3 jcm-11-03324-t003:** Comparison of the postoperative radiographic parameters between the anatomical arthroplasties performed via a deltopectoral approach and a postero-inferior approach.

Measurement Parameter	Deltopectoral (*n* = 9)	Postero-Inferior (*n* = 9)	*p*-Value
Neck-Shaft Angle post-operative (°)	127 ± 4 (range 121 to 134)	130 ± 8 (range 120–143)	0.566
COR medio-lateral (%)	−0.2 ± 3.6 (range −6.9 to 4.1)	3.7 ± 3.4 (range −2.3 to 8.7)	**0.034**
COR supero-inferior (%)	1.0 ± 2.1 (range −1.8 to 4.0)	2.6 ± 2.0 (range 0.6 to 5.4)	0.354
COR antero-posterior (%)	1.7 ± 1.5 (range −0.1 to 4.4)	0.9 ± 2.1 (range −3.9 to 3.8)	0.566
Head size supero-inferior (%)	98.5 ± 0.9 (range 96.5 to 99.6)	97.3 ± 2.6 (range 92.4 to 101.8)	0.145
Head size antero-posterior (%)	100.6 ± 2.1 (range 97.2 to 103.9)	101.1 ± 2.2 (range 98.4 to 105.6)	0.536
Anatomic Reconstruction Score	9.3 ± 1.1 (range 7 to 10)	8.4 ± 1.2 (range 7 to 10)	0.129
Glenoid positioning supero-inferior (%)	51.1 ± 3.9 (range 44.6 to 56.1)	49.1 ± 4.6 (range 40.4 to 57.7)	0.331
Glenoid positioning antero-posterior (%)	50.4 ± 1.4 (range 48.8 to 52.4)	49.3 ± 2.1 (range 44.3 to 51.6)	0.233
Theoretical glenoid inclination (°)	88 ± 4 (range 83 to 95)	86 ± 6 (range 78 to 94)	0.270
Theoretical glenoid retroversion (°)	89 ± 2 (range 86 to 93)	91 ± 6 (range 82 to 103)	0.269

## Data Availability

Data available on request due to restrictions, e.g., privacy or ethical.

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
