# Peer review of "Implant Sizing and Positioning in Anatomical Total Shoulder Arthroplasty Using a Rotator Cuff-Sparing Postero-Inferior Approach"

_jcm, 2022, doi:10.3390/jcm11123324_

Round 1

Reviewer 1 Report

Dear Moroder et al.,

The manuscript “Implant Sizing and Positioning in Anatomical Total Shoulder Arthroplasty using a Rotator Cuff-sparing Postero-inferior Approach” (jcm-1689298) by Moroder et al. compare the effectiveness of a rotator cuff-sparing postero-inferior (PI) approach with subdeltoidal access to the traditional subscapularis-takedown deltopectoral approach in terms of implant sizing and positioning in anatomical total shoulder arthroplasty (aTSA). The topic is interesting, but I think this article should be reconsidered after proper changes in major revision. Some of my specific comments are:

  1. Describe the novelty of the article made by the author? From the results of my evaluation, it seems that many similar published works adequately explain what you have raised in the current manuscript related to Rotator Cuff-sparing Total Shoulder Arthroplasty as the best reviewer knowledge in this research area. In addition, there has been an identical publication by the author (Implant Sizing And Positioning In Rotator Cuff-Sparing Postero-Inferior Approach Anatomical Total Shoulder Arthroplasty, https://doi.org/10.1016/j.jse.2022.01.006) which makes originality and the freshness of the current work is questioned and causes it to be rejected and not worthy of publication. If there is something others really new in this manuscript, please highlight it more clearly in the introduction section (line 40-88).
  2. The state of the art and the significance of the present study are not clearly present, the authors should highlight it more advanced in the introduction section (line 40-88).
  3. In the introduction section, the authors should explain the previous research conducted and its shortcomings. It will uphold the research gap that you filled with your research novelty. I recommend the authors elaborate on their introduction section (line 40-88). Do not forget to attention carefully to my previous comments on numbers 1 and 2.
  4. Since the present study investigate joint arthroplasty, I would encourage and advise the authors to adopt some of the specific additional references related to joint arthroplasty published by MDPI in the introduction section (line 40-88) as follow:
    • Tresca Stress Simulation of Metal-on-Metal Total Hip Arthroplasty during Normal Walking Activity. Materials (Basel). 2021, 14, 7554. https://doi.org/10.3390/ma14247554
    • The Effect of Bottom Profile Dimples on the Femoral Head on Wear in Metal-on-Metal Total Hip Arthroplasty. J. Funct. Biomater. 2021, 12, 38. https://doi.org/10.3390/jfb12020038
  1. In the materials and methods section (line 108-370), the authors should add one systematic figure to illustrate the workflow of experimental testing in the present study to make the reader more interested and easier to understand rather than only using dominant text to explain.
  2. The author must provide a detailed specification and use condition more detail regarding all tools used in the research carried out in the Materials and Methods section (line 108-370) so that the reader can estimate the accuracy and differences in the results that the authors describe due to the use of different tools in future studies.
  3. In the Results section (line 242-282), authors are advised to compare the results they obtain with previous similar/identical studies if it is possible.
  4. Is there any limitation that can be explained by the authors other than stated in the present form (line 358-371), I think it is some that not mentioned, please extend this information.
  5. The conclusion (line 372-379) of the present manuscript is not solid. Further elaboration is needed.
  6. Further research needs to be explained in the conclusion section (line 372-379).
  7. In some place of the manuscript, the authors sometimes made a paragraph only consisting of one or two sentences that made the explanation not clearly understood. The authors need to extend their explanation to become a more comprehensive paragraph. In one paragraph, it is recommended to consist of at least 3 sentences with 1 sentence as the main sentence and the other sentences as supporting sentences. Or the authors can arrange it with combine from the others paragraph. For example in line 90-91.
  8. It seems that in the present manuscript, the authors do not follow Journal of Clinical Medicine, MDPI manuscript template properly and have a misunderstanding using the manuscript template. The authors can download published manuscripts by Journal of Clinical Medicine, MDPI, and compare them with the present author's manuscript to ensure typesetting is appropriate. Some of them that are:
  • Email in each authors do not provided in the affiliation information (line 7-12)
  • Acknowledgments information that should be present before references and conclusion is missing
  • Uppercase and lowercase for all of Subsection is error
  • Reference typesetting format is not correct
  • And others
  1. To improve the quality of English used in this manuscript and make sure English language, grammar, punctuation, spelling, and overall style are correct, further proofreading is needed. As an alternative, the authors can use the MDPI English proofreading service for this issue.

I am pleased to have been able to review the author's present manuscript. Hopefully, the author can revise the current manuscript as well as possible so that it becomes even better. Good luck for the author's work and effort.

Best regards,

The Reviewer

Author Response

Reviewer 1:

COMMENT: Describe the novelty of the article made by the author? From the results of my evaluation, it seems that many similar published works adequately explain what you have raised in the current manuscript related to Rotator Cuff-sparing Total Shoulder Arthroplasty as the best reviewer knowledge in this research area. In addition, there has been an identical publication by the author (Implant Sizing And Positioning In Rotator Cuff-Sparing Postero-Inferior Approach Anatomical Total Shoulder Arthroplasty, https://doi.org/10.1016/j.jse.2022.01.006) which makes originality and the freshness of the current work is questioned and causes it to be rejected and not worthy of publication. If there is something others really new in this manuscript, please highlight it more clearly in the introduction section (line 40-88).

REPLY: We thank the reviewer for this comment. However, the mentioned publication is only an abstract and not an original article. This manuscript describes original work and describes the mentioned PI approach for the first time in literature. Furthermore, it is not under consideration by any other journal.

COMMENT: The state of the art and the significance of the present study are not clearly present, the authors should highlight it more advanced in the introduction section (line 40-88).

REPLY:  Yes, the joint, which was treated at our hospital had at least one previous septic surgery previously. As the number of the patients is to low, no statistically reasonable conclusion can be made regarding groups with and without previous septic surgery.

COMMENT: In the introduction section, the authors should explain the previous research conducted and its shortcomings. It will uphold the research gap that you filled with your research novelty. I recommend the authors elaborate on their introduction section (line 40-88). Do not forget to attention carefully to my previous comments on numbers 1 and 2.

REPLY: We thank for this comment. The previous research about different rotator cuff sparing approaches were discussed between line 54 and 81. These approaches include an anterior deltopectoral approach with only partial take-down of the inferior subscapularis, a superior approach through the deltoid and the rotator interval, and an anterior deltopectoral approach through the rotator interval. Furthermore, Gagey et al. described a posterolateral transdeltoid approach with osteotomy of the external rotators which allows a wide exposure suitable for primary or revision total shoulder arthroplasty and Greiwe et al. reported 6-months results for aTSA implanted using a transdeltoid posterior approach with rotator cuff-sparing internervous access to the joint between the infraspinatus and the teres minor with an in-situ osteotomy of the humeral head. However, the authors also stress the point that it is a challenging technique which should not be attempted in clinical practice without proper training. Furthermore in-situ osteotomy without dislocation of the humeral head can lead to higher failure rate in humeral head osteotomy. In contrast to these posterior approaches, we describe a new PI approach with subdeltoidal access and posterior dislocation of the humeral head through an internervous split between the infraspinatus and the teres minor with medial T-shaped incision of the capsule allowing for better visualization and orientation to perform rotator-cuff sparing anatomical total shoulder arthroplasty.

COMMENT: Since the present study investigate joint arthroplasty, I would encourage and advise the authors to adopt some of the specific additional references related to joint arthroplasty published by MDPI in the introduction section (line 40-88) as follow:

REPLY: We thank the reviewer and added both researches in our manuscript.

COMMENT: In the materials and methods section (line 108-370), the authors should add one systematic figure to illustrate the workflow of experimental testing in the present study to make the reader more interested and easier to understand rather than only using dominant text to explain.

 REPLY: We thank the reviewer for this comment. As the reviewer mentions right, it is important to add figures to illustrate the workflow. Therefore, we added pictures of the procedure of the approach. Other than that, the other surgical steps are usual steps used in regular shoulder arthroplasty.

COMMENT: The author must provide a detailed specification and use condition more detail regarding all tools used in the research carried out in the Materials and Methods section (line 108-370) so that the reader can estimate the accuracy and differences in the results that the authors describe due to the use of different tools in future studies.

REPLY: That’s a very important point. However, the approach is briefly described in our methods section, which makes our study reproducible.

COMMENT: Is there any limitation that can be explained by the authors other than stated in the present form (line 358-371), I think it is some that not mentioned, please extend this information.

REPLY: We believe, that all the limitations are added to the manuscript including the most important one that the implantation of arthroplasties in cadaveric shoulders is typically easier due to the reduced tension of the soft tissues, which can impair our results.

COMMENT: Further research needs to be explained in the conclusion section (line 372-379).

REPLY: Further research should focus on the radiological and clinical outcomes of the PI approach in daily practice.

COMMENT: It seems that in the present manuscript, the authors do not follow Journal of Clinical Medicine, MDPI manuscript template properly and have a misunderstanding using the manuscript template. The authors can download published manuscripts by Journal of Clinical Medicine, MDPI, and compare them with the present author's manuscript to ensure typesetting is appropriate.

REPLY: We corrected the missing parts of our manuscript according to the MDPI manuscript template.

Reviewer 2 Report

The authors aim to compare the PI and deltopectoral approach in TSA.

The authors must be congratulated for this interesting work but also for the quality of the article.

Methodology of the study is almost perfect except the need to change the T-shaped capsular incision that could have had a small influence on the results.

I have few questions authors should address.

Line 118: how was performed the evaluation ? Quantitative/Subjective?

Line 124: in which position was set the cadaver for the PI approach ?

Line 127: posterior border of the acromion? At which level ?

Line 131: the separation between both muscles could be difficult to identify. Do the author have any tricks to share to help in this step.

Line 155: the questions should be provided at the Methods

Line 175: “where the deviation between pre- and postoperative COR could be determined” not very clear. Please specify.

Results

Line 271-280: redundant with the table 3, should be removed

Line 299: can be confusing. “PI had a more medialized prosthetic COR than in the deltopectoral approach”

Discussion

Line 322-325: indeed, those are the two main issues of the PI approach that deserve to be more deeply studied.

Author Response

COMMENT: Line 124: in which position was set the cadaver for the PI approach ?

REPLY: The cadavers were positioned in a beach chair position

COMMENT: Line 127: posterior border of the acromion? At which level ?

REPLY: We changed the following statement in our manuscript: A 10-12 cm vertical skin incision was made on the posterior aspect of the shoulder beginning at the posterior border of the acromion around 2 cm medial of the lateral aspect.

COMMENT: Line 131: the separation between both muscles could be difficult to identify. Do the author have any tricks to share to help in this step.

REPLY: That is very correct. But in 2/3 of the specimen we could have identify a fat pad between infraspinatus and teres minor. We added the following statement in our manuscript: A fat line between infraspinatus and teres minor could have been identified in two-thirds of the specimen to aid in identifying the internervous interval. Line 136-137

REPLY: Line 155: the questions should be provided at the Methods

COMMENT: we changed the statement in our manuscript as follows: While performing the procedure the surgeon had to grade the the difficulty (poor, acceptable or excellent) to achieve a certain surgical step including identification of the internervous interval between infraspinatus and teres minor, exposure of the humeral head, humeral head resection, and exposure of the glenoid as well as glenoid reaming.

Line 160-163

COMMENT: Line 175: “where the deviation between pre- and postoperative COR could be determined” not very clear. Please specify.

REPLY: We changed the statement in our manuscript as follows: The deviation between pre- and postoperative COR was then determined in the x and y axis. Line 181-182

COMMENT: Line 271-280: redundant with the table 3, should be removed

REPLY: We removed the redundant information from the results section.

COMMENT: Line 299: can be confusing. “PI had a more medialized prosthetic COR than in the deltopectoral approach”

REPLY: we change the statement in our manuscript as followed: The COR was slightly but statistically significantly more medial in cases with PI approach than with the deltopectoral approach indicating a risk of lateral overstuffing due to insufficient resection of the humeral head. Line 320-321

COMMEENT: Discussion

Line 322-325: indeed, those are the two main issues of the PI approach that deserve to be more deeply studied.

REPLY: We agree with the reviewer and further human studies should focus whether posterior approaches may also weaken the posterior soft tissues, including posterior capsule and rotator cuff, and therefore possibly aggravate posterior humeral subluxation in patients with posterior eccentric glenoid wear.

Round 2

Reviewer 1 Report

The authors have revised the manuscript and addressed all of my comments. I am recommending this manuscript be accepted.